# MicroRNA Signatures in Cartilage Ageing and Osteoarthritis

**DOI:** 10.3390/biomedicines11041189

**Published:** 2023-04-17

**Authors:** Panagiotis Balaskas, Katarzyna Goljanek-Whysall, Peter D. Clegg, Yongxiang Fang, Andy Cremers, Aibek Smagul, Tim J. M. Welting, Mandy J. Peffers

**Affiliations:** 1Institute of Life Course and Medical Sciences, William Henry Duncan Building, 6 West Derby Street, Liverpool L7 8TX, UK; 2Department of Physiology, College of Medicine, Nursing and Health Sciences, University of Galway, H91 TK33 Galway, Ireland; 3Centre for Genomic Research, Institute of Integrative Biology, Biosciences Building, University of Liverpool, Crown Street, Liverpool L69 7ZB, UK; 4Department of Orthopaedic Surgery, Medical Centre, Maastricht University, 6202 AZ Maastricht, The Netherlands

**Keywords:** osteoarthritis, knee cartilage, microarray, miR-107, miR-143-3p

## Abstract

Osteoarthritis is the most common degenerative joint disorder. MicroRNAs are gene expression regulators that act post-transcriptionally to control tissue homeostasis. Microarray analysis was undertaken in osteoarthritic intact, lesioned and young intact cartilage. Principal component analysis showed that young intact cartilage samples were clustered together; osteoarthritic samples had a wider distribution; and osteoarthritic intact samples were separated into two subgroups, osteoarthritic-Intact-1 and osteoarthritic-Intact-2. We identified 318 differentially expressed microRNAs between young intact and osteoarthritic lesioned cartilage, 477 between young intact and osteoarthritic-Intact-1 cartilage and 332 between young intact and osteoarthritic-Intact-2 cartilage samples. For a selected list of differentially expressed microRNAs, results were verified in additional cartilage samples using qPCR. Of the validated DE microRNAs, four—miR-107, miR-143-3p, miR-361-5p and miR-379-5p—were selected for further experiments in human primary chondrocytes treated with IL-1β. Expression of these microRNAs decreased in human primary chondrocytes treated with IL-1β. For miR-107 and miR-143-3p, gain- and loss-of-function approaches were undertaken and associated target genes and molecular pathways were investigated using qPCR and mass spectrometry proteomics. Analyses showed that *WNT4* and *IHH*, predicted targets of miR-107, had increased expression in osteoarthritic cartilage compared to young intact cartilage and in primary chondrocytes treated with miR-107 inhibitor, and decreased expression in primary chondrocytes treated with miR-107 mimic, suggesting a role of miR-107 in chondrocyte survival and proliferation. In addition, we identified an association between miR-143-3p and EIF2 signalling and cell survival. Our work supports the role of miR-107 and miR-143-3p in important chondrocyte mechanisms regulating proliferation, hypertrophy and protein translation.

## 1. Introduction

Osteoarthritis (OA) is a joint disease most commonly affecting the knee, hip, hand, foot and the spine. It is one of the leading causes of disability, and millions are affected worldwide; approximately 250 million people are suffering from knee OA alone [1]. It is now accepted that OA is a disease of the whole joint with several joint tissues affected [2]. Cartilage loss is a major clinical feature and patients with end-stage OA usually undergo total knee replacement surgery [3].

There are many risk factors associated with the initiation and progression of knee OA; systemic risk factors such as age, genetics, sex, and ethnicity and mechanical risk factors which include joint structure, joint trauma and physical activity [1,2]. Ageing is the most common risk factor leading to structural and molecular changes in articular cartilage, making it susceptible to OA [4,5]. These changes include reduced synthesis of anabolic markers and extracellular matrix components [5], collagen cross-linking [4], increased chondrocyte senescence [5], oxidative stress [6,7] and increased joint inflammation [8].

MicroRNAs (miRNAs or miRs) are short non-coding RNAs between 17 and 25 nucleotides long. They are epigenetic, post-transcriptional regulators of gene expression and function, targeting complementary sequences in the 3′ untranslated region (UTR) of their mRNA targets, thus leading to translational repression and/or mRNA decay [9]. Biogenesis of canonical miRNAs is a complex process that involves several steps. In the nucleus, miRNAs genes are first transcribed into long primary miRNA (pri-miRNA) transcripts, which are double-stranded stems with hairpin structures. Pri-miRNAs undergo cleavage by the microprocessor complex which consists of two proteins: Drosha, a ribonuclease III (RNase III) enzyme, and DiGeorge critical region 8 (DGCR8) [9,10]. This gives rise to the precursor-miRNA (pre-miRNA), which is a ~70 nt stem-loop structure. Pre-miRNAs exit the nucleus and are exported to the cytoplasm via exportin-5. There, they are cleaved further by another endoribonucleolytic enzyme with RNase III activity, Dicer [10]. After cleavage, there is a ~22 nt RNA duplex left. This RNA duplex is loaded into an Argonaute (Ago) protein to form the RNA-induced silencing complex (RISC). One strand, called the guide strand, is stabilised within the complex, while the remaining strand, called the passenger strand, is degraded [9,10,11]. Mature miRNAs exert their function by pairing with their mRNA targets. Nucleotides 2 to 7 or 2 to 8 relative to the 5′ end of the mature miRNA are called the ‘seed’ region and are the most important for miRNA–mRNA pairing [12]. In contrast to plants, where there is a near perfect complementarity between the miRNA seed region and the 3′ UTR of the mRNA target, in animals there is only partial complementarity. This means that a single miRNA can bind different mRNA targets [12]. There are two proposed ways of miRNA-induced silencing of gene expression in animals: translational repression and mRNA decay. Both mechanisms include a plethora of molecular components, such as trinucleotide repeat containing adaptor 6 (TNRC6), decapping factors and deadenylase complexes [12,13]. The relative contribution of each mechanism is still unclear, however, the current consensus is that these two mechanisms work together to induce gene silencing and are closely linked [12].

The miRNAs play an important role in the skeletal system including cartilage and bone development by regulating key processes, such as chondrocyte proliferation, differentiation and apoptosis [14]. Chondrocyte-specific deletion of DICER in the growth plate of mice reduced the number of proliferating chondrocytes, increased hypertrophic chondrocytes and led to severe skeletal defects and premature death [15]. Several miRNAs have been associated with cartilage development and homeostasis and dysregulation of these is linked with ageing and OA [16,17,18,19]. miR-140 is one of the most important miRNAs in cartilage. Expression of miR-140 was reduced in cartilage progenitors/stem cells derived from OA cartilage, whereas miR-140 mimics alleviated OA-like changes in these stem cells [20]. Treatment of chondrocytes with exosomes overexpressing miR-140 resulted in increased secretion of extracellular matrix components, such as collagen and aggrecan, and in addition, intra-articular injection of these exosomes resulted in increased cartilage regeneration in an OA rat model [21]. miR-140 expression is regulated by *SOX9*, the master transcription regulator of chondrogenesis (Miyaki et al., 2010) and decreased miR-140 expression has been correlated with reduced *SOX9* and aggrecan levels [22]. Two other miRNAs studied in OA are miR-181 and miR-455. miR-181 expression increased in a rat OA model and treatment with miR-181 mimics resulted in elevated inflammatory response, whereas treatment with miR-181 inhibitor had the opposite effect [23]. miR-455 had lower expression in OA human cartilage and miR-455 knock-out mice presented with OA-like features via upregulation of the hypoxia inducible factor-2α (HIF-2α), a cartilage catabolic marker, which is a direct target of miR-455 [24]. A number of other miRNAs have been linked to OA development and several OA-related cellular processes, such as inflammation, autophagy, senescence and oxidative stress [25]. Inflammatory cytokines, such as interleukin 1 beta (IL-1β), exert catabolic effects in cartilage and promote degradation. Synovial tissues from OA patients are characterised by distinctive changes in the expression of several miRNAs, including miR-146a and miR-335, and these changes suggest a role of miRNAs in OA inflammation [18]. Moreover, a recent study by Zhang et al. (2023) demonstrated aberrant miRNA expression in senescent chondrocytes with 279 miRNAs differentially expressed between senescent and non-senescent chondrocytes [26]. In the same study, miR-135b was identified as a promising regulator of chondrocyte senescence, and reduced miR-135b expression was linked to increased senescence [26]. miR-525 is another miRNA linked to oxidative stress in chondrocytes. miR-525 interacts with CircFNDC3B, a circular RNA which exerts an anabolic effect in cartilage by promoting cell proliferation and decreasing the level of reactive oxygen species in chondrocytes [26]. Furthermore, some miRNAs have been associated with OA through their role in modulating autophagy in chondrocytes. Increased levels of miR-378 repressed autophagy and induced inflammation and chondrocyte hypertrophy by binding *Atg2a*, a gene which promotes autophagy [27]. The involvement of miRNAs in important cartilage- and OA-related pathways highlights their potential use as therapeutic agents and disease biomarkers.

Previous work from our group has identified differentially expressed (DE) miRNAs in cartilage ageing and OA [28,29,30]. In this study, we utilised microarray analysis to identify DE miRNAs between young intact knee cartilage from patients undergoing anterior cruciate ligament (ACL) repair surgery, and knee OA intact and lesioned cartilage from patients following total knee arthroplasty (TKA).

## 2. Materials and Methods

### 2.1. Cartilage Tissue Samples

For microarray analysis, full thickness intact cartilage from young patients (*n* = 9, age ± SD = 23.8 ± 3.8), undergoing ACL reconstruction, was collected from the femoral intercondylar notch. In addition, old OA intact and lesioned femoral cartilage (*n* = 10, age ± SD = 62.6 ± 7.3) was collected from OA patients undergoing TKA. Diagnosis of OA was confirmed on preoperative knee radiographs following Kellgren–Lawrence scoring. OA intact and lesioned cartilage was collected from a relatively intact (lateral condyle) and damaged area (medial condyle) in the femoral condyles of the same patients, respectively. For qPCR validation, an additional cohort of cartilage samples was used consisting of full thickness intact cartilage from young patients (*n* = 8, age ± SD = 21 ± 6.5), undergoing ACL reconstruction and OA intact and lesioned cartilage from patients undergoing TKA (*n* = 7, age ± SD = 62.1 ± 3.7).

### 2.2. Isolation of Human Primary Chondrocytes from OA Articular Cartilage

For IL-1β treatment and miRNA mimic/inhibitor treatment, human primary chondrocytes were isolated from the knee articular cartilage of OA patients undergoing TKA, Specifically, under sterile conditions, cartilage tissue was washed with Dulbecco’s phosphate-buffered saline (DPBS) (Sigma-Aldrich, Dorset, UK) and cut into 2–3 mm pieces. Cartilage pieces were placed in 50 mL of Dulbecco’s Modified Eagle Medium (DMEM) (ThermoFisher Scientific, Paisley, UK) supplemented with 10% foetal bovine serum (FBS), 1% penicillin-streptomycin (P/S), 0.2% amphotericin B/fungizone (F/Z) (Gibco/ThermoFisher Scientific, Paisley, UK) and 0.08% collagenase type 2 (Worthington Biochemicals, Danehill, UK) and placed in a shaking incubator at 37 °C overnight for tissue digestion and chondrocyte extraction. Next day, the solution was filtered through a 70 μm sterile cell strainer (Fisher Scientific, Loughborough, UK) and centrifuged at 1400 rounds per minute (rpm) at room temperature for 4 min. Supernatant was removed and chondrocytes were resuspended in 12 mL complete media and transferred to a T75 flask (Greiner Bio-One, Stonehouse, UK) to attach. Flask was placed in a tissue culture incubator at 37 °C, supplied with 20% O_2_ and 5% CO_2_ until cells reached 80% confluence.

### 2.3. Histology

To microscopically characterise intact and lesioned OA human cartilage collected for microarray analysis, cartilage tissue for histological assessment was collected from the same OA donors used for microarray analysis and processed as previously described [31]. Briefly, tissue was fixed in 4% paraformaldehyde and embedded in paraffin wax. Tissue was sectioned at 4 μm using a Leica RM 2235 microtome (Leica Biosystems, Milton Keynes, UK). Sections were placed on microscope slides (ThermoFisher Scientific, Paisley, UK) and left to dry on a warm surface for 15 min and then at 37 °C overnight. For Haematoxylin and Eosin staining, sections were dewaxed in consecutive solutions of xylene and ethanol (Fisher Scientific, Loughborough, UK) of decreasing concentration (100%-90%-70% *v*/*v*). Sections were stained with Harris haematoxylin (Leica Biosystems, Milton Keynes, UK) for 7 min. Sections were counterstained with alcoholic eosin (Leica Biosystems, Milton Keynes, UK) for 10 min. Sections were dehydrated by placing them in consecutive solutions of increasing ethanol concentration (96–100% *v*/*v*) and xylene. For Safranin-O/Fast Green staining, the same dewaxing process was used as above. Then, sections were placed in Harris haematoxylin for 30 s, and then stained with 0.2% (*w*/*v*) fast-green solution (Sigma-Aldrich, Dorset, UK) for 10 min. Sections were washed shortly with water for 10 s, then stained with 0.5% (*w*/*v*) safranin-O solution (Sigma-Aldrich, Dorset, UK) for 2 min and dehydrated as above. Sections were viewed using a Nikon Eclipse 80i microscope (Nikon, UK) and scored using the Mankin histological–histochemical grading system (HHGS) [32] by two independent scorers.

### 2.4. RNA Isolation

For microarray analysis, approximately 100 mg of human cartilage tissue per sample was pulverized under liquid nitrogen using a mikro-dismembrator S (Sartorius, Melsungen, Germany) and total RNA was isolated using the mirVana™ miRNA Isolation Kit with phenol (Life Technologies, Paisley, UK), according to the manufacturer’s protocol.

For qPCR analysis, total RNA was extracted from OA primary chondrocytes using the guanidinium thiocyanate-phenol-chloroform method [33]. RNA concentration and purity were measured using a Nanodrop 2000 Spectrophotometer (NanoDrop Technologies, Wilmington, NC, USA).

### 2.5. Microarrays and Data Analysis

Microarray analysis was undertaken using the Affymetrix GeneChip^®^ miRNA 4.0 Arrays (ThermoFisher Scientific, Paisley, UK). Arrays assayed for all mature miRNAs in miRbase Release 20 and contained probes for 2578 human mature miRNAs and 2025 human precursor miRNAs. For hybridisation, 500 ng of total RNA per sample was prepared using the Affymetrix FlashTag™ Biotin HSR RNA Labeling Kit (ThermoFisher Scientific, Paisley, UK), according to manufacturer’s instructions. After hybridisation, the hybridisation cocktail was extracted from each array and the arrays were washed using the Affymetrix Hybridisation wash and stain kit on the GeneChip Fluidics station 450 (ThermoFisher Scientific, Paisley, UK) using fluidics script FS450_0002, and scanned with the Affymetrix GeneChip scanner 3000 7G (ThermoFisher Scientific, Paisley, UK).

After scanning the arrays, Affymetrix array .CEL files were generated using the Affymetrix GeneChip Command Console Software. Data were pre-processed using Robust Microarray data Analysis (RMA) to obtain normalised small RNA expression data. Model-based analysis of small RNA expression was applied to data pre-processed with RMA which generated normalised expression values presented on a log2 scale. Data quality were assessed by measuring small RNA expression distribution. The random variation of the data were formulated following a normal distribution, and the model was linear, taking the mean of each group as a model parameter. The log2 Fold Change (logFC) was computed from model fitting results and tested using *t*-tests to get associated *p*-values. *p*-values were adjusted for multiple testing using the False Discovery Rate (FDR) approach [34]. Significant DE miRNAs were defined as those with FDR-adjusted *p*-value < 0.05. All the processes were performed in R environment using limma package [35].

### 2.6. cDNA/Poly(A) cDNA Synthesis for mRNA/miRNA Quantification

For miRNA quantification, 200 ng of total RNA were converted into poly(A) cDNA using the miScript II RT Kit (Qiagen, Manchester, UK), according to the manufacturer’s protocol. Final reaction volume was 20 μL. For mRNA quantification, 500 ng of total RNA were converted into cDNA using the M-MLV reverse transcriptase (Promega, Southampton, UK), according to the manufacturer’s protocol. Final reaction volume was 25 μL.

### 2.7. qPCR for miRNA/mRNA Quantification

MiRNA amplification by qPCR was undertaken using the miScript SYBR^®^ Green PCR Kit and the appropriate miScript Primer Assay (Qiagen, Manchester, UK) according to the manufacturer’s protocol. Per reaction, 5 ng of poly(A) cDNA was mixed with 7.5 μL of QuantiTect SYBR Green PCR Master Mix, miRNA-specific forward primer and a universal reverse primer at a final concentration of 330 nM per primer. Nuclease-free water was added to bring the reaction to a final volume of 15 μL. miRNA amplification was undertaken using a LightCycler^®^ 96 Instrument (Roche Life Science, Penzberg, Germany) under the following cycling conditions: 1 cycle of: 95 °C for 15 min, then 45 cycles of: 94 °C for 15 s, 55 °C for 30 s, and 70 °C for 30 s. For cartilage tissue, relative expression levels were normalised to miR-6786-5p which had stable expression in all cartilage samples according to microarray data. For primary chondrocytes, relative expression levels were normalised to SNORD68 which, from our experience, had the most stable expression in chondrocytes. Relative expression was calculated using the 2^−ΔCt^ method [36]. MiRNA primers were manufactured by Qiagen and the sequence is proprietary to Qiagen.

mRNA quantification by qPCR was undertaken using the QuantiTect SYBR Green PCR Master Mix (Qiagen, Manchester, UK), according to the manufacturer’s protocol. Per reaction, 10 ng of cDNA was mixed with 5 μL of QuantiTect SYBR Green PCR Master Mix and gene-specific forward and reverse primer mix at a final concentration of 400 nM per primer. Nuclease-free water was added to bring the reaction to a final volume of 10 μL. Cycling conditions were as above. For cartilage tissue, relative expression levels were normalised to Ribosomal Protein L3 (*RPL3*). For extracted primary chondrocytes relative expression levels were normalised to Glyceraldehyde 3-phosphate dehydrogenase (*GAPDH*). In our hands, these reference genes have the most stable expression in cartilage tissue and chondrocytes, respectively. Relative expression was calculated using the 2^−ΔCt^ method [36]. Protein coding gene primers were manufactured by either Eurogentec or Primerdesign (Appendix A).

### 2.8. Treatment of Human Primary Chondrocytes with IL-1β

To investigate miRNA expression in an in vitro inflammatory model of OA, chondrocytes extracted from OA donors (*n* = 4–5 donors per miRNA) undergoing TKA surgery were treated with 10 ng/mL IL-1β. Cells were grown to passage 1 at 80% confluence, then seeded in well-plates at a seeding density of 30,000 cell/cm^2^ in DMEM supplemented with 10% FBS, 1% penicillin-streptomycin and 0.2% amphotericin B/fungizone at 37 °C, with 20% (*v*/*v*) O_2_ and 5% (*v*/*v*) CO_2_ overnight. Next day, chondrocytes were placed in DMEM supplemented with 1% P/S and 0.2% F/Z but no FBS and serum-starved overnight. The following day, cells were treated with FBS-free DMEM supplemented with 10 ng/mL IL-1β (R&D Systems, Abingdon, UK) or phosphate-buffered saline (PBS)/BSA as control for 24 h or 5 days. At each timepoint, cells were washed with PBS and lysed and total RNA was extracted for miRNA/mRNA amplification as described above.

### 2.9. Overexpression and Knockdown of miRNAs in IL-1β Induced Human Primary Chondrocytes

MiRNA mimics and inhibitors were used to overexpress and knockdown miR-107 and miR-143-3p in IL-1β induced human primary chondrocytes. Two commercially available systems were purchased from Qiagen to undertake these experiments. For miR-107, the miScript mimics, inhibitors and controls were used (Syn-hsa-miR-107 miScript miRNA Mimic/AllStars Negative Control siRNA and Anti-hsa-miR-107 miScript miRNA Inhibitor/miScript Inhibitor Negative Control). For miR-143-3p, the miRCURY LNA mimic, inhibitor and controls were used (hsa-miR-143-3p miRCURY LNA miRNA Mimic/Negative Control miRCURY LNA miRNA Mimic and hsa-miR-143-3p miRCURY LNA miRNA Inhibitor/miRCURY LNA miRNA Inhibitor Negative Control A). Briefly, chondrocytes extracted from OA donors (*n* = 8–11 donors) undergoing TKA surgery were grown to passage 1 and then seeded in well-plated in DMEM supplemented with 10% FBS, 1% P/S and 0.2% F/Z at a density of 25,000–30,000 cells/cm^2^. Cells were left to attach and proliferate overnight. Next day, chondrocytes were placed in FBS-free DMEM media supplemented with 10 ng/mL IL-1β. Mimic, inhibitor and respective controls were prepared according to manufacturer’s recommendations. For miR-107, cells were treated to a final mimic/control concentration of 5 nM and a final inhibitor/control concentration of 200 nM for 48 h. For miR-143-3p, cells were treated to a final mimic/control concentration of 0.5 nM and a final inhibitor/control concentration of 50 nM for 48 h. Following treatment, cells treated with miR-107 mimic/inhibitor/controls were collected for qPCR analysis, and cells treated with miR-143-3p mimic/inhibitor/controls were collected for liquid chromatography tandem mass spectrometry (LC-MS/MS) analysis.

### 2.10. Protein Extraction, In-Solution Digestion and LC-MS/MS Following Overexpression and Knockdown

To investigate the targetome of miR-143-3p, chondrocytes from OA donors (*n* = 5 donors) undergoing TKA surgery treated with miR-143-3p mimics and inhibitors were collected and prepared for LC-MS/MS. Specifically, cells were collected in 25 mM ammonium bicarbonate (AmBic) (Sigma-Aldrich, Dorset, UK) prepared in Pierce™ Water, LC-MS Grade and supplemented with 1× protease inhibitor cocktail (Roche Life Science, Penzberg, Germany). 7.5 units of benzonase nuclease (Biocompare, San Francisco, CA, USA) was added per sample and mixed slightly to release histones from DNA. Cells were lysed by sonication using a Soniprep 150 ultrasonic disintegrator (MSE, East Sussex, UK). Cells were sonicated on ice, in three rounds of 30 s each, with 1 min and 15 s rest between each round. Volume was adjusted to 200 μL by adding 25 mM AmBic supplemented with 1x protease inhibitor cocktail. Per sample 20 μL was taken for protein quantification using the Pierce™ 660 nm Protein Assay (ThermoFisher Scientific, Paisley, UK) according to the manufacturer’s protocol. A total of 10 μL was taken for protein quality analysis by Silver Staining. Briefly, following protein quantification, 2 μg of protein per sample was subjected to sodium dodecyl sulphate–polyacrylamide gel electrophoresis (SDS-PAGE) using the NuPAGE™ 4% to 12% (*w*/*v*), Bis–Tris gel system (ThermoFisher Scientific, Paisley, UK) and according to the manufacturer’s recommendations. Following SDS-PAGE, protein samples were subjected to Silver Staining using the Pierce Silver Stain Kit (ThermoFisher Scientific, Paisley, UK) according to manufacturer’s protocol. The gel was visualised using the ChemiDoc XRS+ Gel Imaging System (Bio-Rad, Watford, UK).

For in-solution tryptic digestion, 35 μg of protein in 160 μL 25 mM AmBic starting volume was used. Specifically, samples were randomised for processing to avoid bias. A total of 10 μL of freshy prepared 1% (*w*/*v*) RapiGest SF Surfactant (Waters, Herts, UK) was added to each sample and samples were incubated at 80 °C for 10 min. Then, 11.1 mg/mL solution of dithiothreitol (DTT) (Sigma-Aldrich, Dorset, UK) was added to each sample at final DTT concentration of 4 mM and samples were incubated at 60 °C for 10 min. Following this, iodoacetamide (IAA) solution (Sigma-Aldrich, Dorset, UK) was added to each sample (9 mM final concentration) and incubated at room temperature in the dark for 30 min. To prevent overalkylation, additional 11.1 mg/mL DTT solution was added to each sample (final DTT concentration 7 mM). For protein digestion, 2 μg of Trypsin/Lys-C Mix, Mass Spec Grade (Promega, Southampton, UK) was added to every sample and samples were rotated at 37 °C for 2 h initially and then treatment was repeated overnight. Next day, trifluoroacetic acid (TFA) (Sigma-Aldrich, Dorset, UK) was added to each sample at a final concentration of 0.5% (*v*/*v*) and samples were incubated at 37 °C for 45 min. Samples were centrifuged at 13,000× *g* at 4 °C for 15 min to remove all insoluble. Supernatant was transferred to new tubes and samples were centrifuged again at 13,000× *g* at 4 °C for 15 min. 1 μg of digested protein was subjected to SDS-PAGE and Coomassie Brilliant Blue staining using the Coomassie Brilliant Blue Staining R-250 Staining Solution (Bio-Rad, Watford, UK) according to manufacturer’s protocol, to confirm complete protein digestion.

Analysis was undertaken using a QExactive HF quadrupole-Orbitrap mass spectrometer (ThermoFisher Scientific, Paisley, UK) coupled with the UltiMate™ 3000 RSLCnano liquid chromatographer (ThermoFisher Scientific, Paisley, UK) using methods previously described [37]. Digested proteins were run on a 90 min gradient with a 30 min run of blanks between samples.

### 2.11. Bioinformatic Analysis of LC-MS/MS data

The raw files of the acquired spectra were aligned by the Progenesis QI for proteomics software (Waters, Manchester, UK). The top five spectra for each feature were exported from Progenesis QI and used for peptide identification with our local Mascot server (Version 2.6.2) searching against the Unihuman Reviewed database, containing 22,640 protein sequences. Search parameters were adjusted to mass tolerance of 10 ppm, fragment mass tolerance of 0.01 Da, one missed cleavage allowed, with carbamidomethyl cysteine as a fixed modification and methionine, proline, lysine oxidation as variable modifications. For statistical analysis, proteins that had at least two unique peptides were selected. To assess differences between mimic and control, and inhibitor and control groups, two-tailed paired t-tests were undertaken using the Stats R package (Version 4.0.2). *p* Values were adjusted for multiple testing. Differences at FDR-adjusted *p* < 0.05 were considered significant.

### 2.12. Functional Enrichment Analysis of miRNA–Gene Interactions

Ingenuity Pathway Analysis (IPA) (Qiagen, Manchester, UK) was used for miRNA pathway analysis and miRNA–mRNA interactions. DE miRNAs and their calculated logFC were uploaded into IPA for target prediction and pathway association filtering for experimentally observed or highly predicted miRNA–mRNA interactions in chondrocyte and cartilage tissue. Predicted mRNAs were re-uploaded into IPA as a new dataset and a Core Analysis was run to identify pathways, diseases, functions and upstream regulators specific to the predicted target genes. To confirm results generated by IPA, lists of all highly predicted mRNA targets from IPA were put into the Enrichr online tool [38,39] for gene enrichment analysis and further pathway identification. Results were listed based on ‘Pathway’, ‘Ontologies’ and ‘Disease/Drugs’ information. To generate and visualise gene networks, datasets of predicted mRNAs were uploaded to ToppGene Suite [40]. This generated a list of gene ontology (GO) terms for biological processes, and each GO term was assigned an FDR value. Go terms with FDR were uploaded onto REVIGO [41] and interactive graphs and tree maps were generated. To better visualise the gene networks generated by REVIGO, these were downloaded and then uploaded on Cytoscape [42]. Within Cytoscape, networks were filtered by ‘value’ and gene interactions were visualised.

### 2.13. STRING Analysis for Identification of Protein–protein Interactions

To identify protein–protein interactions, STRING analysis [43] was undertaken on DE proteins following miR-143-3p experiments. The ‘Multiple proteins’ option was selected and protein lists were uploaded on STRING. Organism was set to ‘Homo sapiens’. Network type was set to ‘full STRING network’ and meaning of network edges was set to ‘evidence’. Minimum required interaction score was set to ‘highest confidence (0.900)’.

### 2.14. Statistical Analysis

Statistical analysis was undertaken in GraphPad Prism version 8.0.1 and normality assessed using the Shapiro–Wilk normality test. For qPCR quantification of miRNAs and mRNAs in human cartilage tissues, the Mann–Whitney test was used as data did not follow the Gaussian distribution and groups were independent. For qPCR quantification of miRNAs and mRNAs in human primary cells treated with IL-1β, miRNA mimics/inhibitors or control, paired t-tests were undertaken between treated and control groups. *p* Values < 0.05 were considered significant. For LC-MS/MS, data variation were assessed using PCA plots generated with MetaboAnalyst 5.0 [44].

## 3. Results

### 3.1. Radiographical and Histological Scoring

The Kellgren–Lawrence grading of lateral and medial femoral condyles of OA patients recruited for microarray along with the histological scores of intact and lesioned human knee cartilage samples from OA patients have been reported previously. Kellgren–Lawrence score of the medial condyles was significantly higher than the lateral condyles (*p* = 0.03). Modified Mankin scores were higher in lesioned cartilage (*p* = 0.04) [31].

### 3.2. Microarray Analysis Revealed Distinctive miRNA Expression between Young Intact and Old OA Human Cartilage

Data quality were assessed by measuring small RNA expression in 9 young intact, 10 old OA intact and 10 old OA lesioned samples. Heatmap of correlation coefficients showed that samples in the young group correlated closer to each other and there was no significant biological variation within the group. In contrast, samples in OA intact and OA lesioned groups were more variable and biological variation was stronger within these two groups (Figure 1A). The 2D PCA plot of the second and third components of log-transformed miRNA abundance showed that young intact cartilage samples (Y) clustered together in one group, separated from OA intact and OA lesioned groups (Figure 1B). OA intact samples were scattered in a wide range and could be divided into two separate subgroups: OA-Intact-1 (samples OA-I-1, OA-I-2, OA-I-3 and OA-I-5) and OA-Intact-2 (samples OA-I-4, OA-I-6, OA-I-7, OA-I-8, OA-I-9 and OA-I-10). OA lesioned samples were also scattered, but the degree of variation was not as high as in the OA intact group.

To investigate possible reasons for clustering within the old OA intact group, we analysed the following available data: age, Kellgren–Lawrence score and Mankin score. There was no significant difference in Kellgren–Lawrence or Mankin scores. There was significant difference (*p* = 0.01) in donor age between OA-Intact-1 (mean age ± standard deviation = 55.5 ± 3.5) and OA-Intact-2 (mean age ± standard deviation = 67.3 ± 4.6). Given no other available clinical data for the donors, we hypothesise that age could account for cluster separation.

To identify DE miRNAs (defined as those with FDR-adjusted *p*-value < 5%), the following comparisons were considered:

OA-Lesioned vs. Y, OA-Intact-1 vs. Y, OA-Intact-2 vs. Y, OA-Lesioned vs. OA-Intact-1, OA-Lesioned vs. OA-Intact-2, and OA-Intact-1 vs. OA-Intact-2. Analysis demonstrated that the list of significant DE miRNAs was similar in the following three comparisons: OA-Lesioned vs. Y, OA-Intact-1 vs. Y and OA-Intact-2 vs. Y, indicating that OA intact and OA lesioned cartilage samples were more similar to each other than to the young cartilage samples. Of the old OA groups, OA-Lesioned and OA-Intact-2 were more similar to each other and less similar when compared to OA-Intact-1 group. Since the biggest differences were noted between the old OA and young intact samples, we focused on the first three comparisons for further analysis. Number of DE miRNAs between these three comparisons are shown in Table 1.

The list of DE miRNAs from these three comparisons were uploaded into IPA for further analysis. MicroRNA Target filter analysis generated a list of experimentally validated and highly predicted mRNA targets. The list was redefined by setting OA- and cartilage-related filters. IPA Core Analysis generated similar results for the three comparisons. Specifically, predicted mRNA targets of DE miRNAs were involved in significant canonical pathways, including osteoarthritis pathway [−log(*p*-value) = 50.04], hepatic fibrosis [−log(*p*-value) = 39.46], role of osteoblasts and chondrocytes in rheumatoid arthritis [−log(*p*-value) = 36.67], IL-17 signalling [−log(*p*-value) = 19.86], HIF1α signalling [−log(*p*-value) = 18.89] and others (Figure 2A). To further investigate the potential pathways that the DE miRNAs and their predicted target genes were associated with, we used the Enrichr online tool [38,39]. The analysis demonstrated that predicted mRNAs of DE miRNAs from OA-Lesioned vs. Y, OA-Intact-1 vs. Y and OA-Intact-2 vs. Y included pathways, such as inflammatory response, cytokine production, cytokine response, regulation of apoptosis and regulation of cell proliferation. To create pathway interactions, gene ontology terms were generated using ToppGene Suite [40] and redefined using Revigo [34] and then visualised using Cytoscape [35]. Gene interaction analysis confirmed results following IPA and Enrichr analyses, that the networks generated included similar pathways (Figure 2B,C).

### 3.3. qPCR Validation of Microarray Findings in Cartilage Samples

For validation of microarray results using qPCR, specific DE miRNAs were selected for follow-up studies. For this research article, we focused on comparisons between OA-Lesioned vs. Y, OA-Intact-1 vs. Y and OA-Intact-2 vs. Y, as the list of DE miRNAs was similar between these and we were interested in including all three major cartilage groups (young, old OA intact and lesioned). Selection was based on fold change, known and/or predicted importance in cartilage biology from the available literature and known/predicted target genes, as well as novelty. For fold change we selected DE miRNAs with logFC < −3 or >3 in all three comparisons. However, none of the DE miRNAs met the logFC > 3 criterion, thus we focused on DE miRNAs with logFC < −3 as the majority of DE miRNAs had significantly lower expression in the OA groups compared to the Young. The following miRNAs were validated: miR-140-5p, miR-155-5p, miR-143-3p, miR-107, miR-379-5p, miR-361-5p and miR-132-5p, and they all had significantly lower expression in the OA groups compared to the Young.

To validate microarray platform, we undertook a qPCR analysis using RNA from the same donors used in the microarrays. Results showed that selected miRNAs had significantly lower expression in the OA-Intact and Lesioned groups when compared to the Young group and followed the same trend as in microarray analysis (Figure 3). For qPCR analysis, OA-Intact-1 and -2 groups were grouped together in one cluster as both groups showed similar direction of dysregulation when compared to the Young group. Moreover, as in microarray analysis, selected miRNAs were not significantly different between OA-Intact and OA-Lesioned groups and their expression was similar between the two groups.

Next, we used an independent cohort of samples to validate our microarray findings. Results indicated that selected miRNAs had significantly lower expression in OA-Intact and Lesioned groups compared to Young intact group. Moreover, as with the dependent cohort, expression of these miRNAs in the OA-Intact group was lower than in the OA-Lesioned group, but this change was not statistically significant (Figure 4).

### 3.4. Investigation of Selected miRNAs in an In Vitro Inflammatory Model of OA

To investigate miRNA response in an OA-like inflammatory system, we treated OA chondrocytes grown in monolayer up to passage 1 with 10 ng/mL IL-1β for 24 h and 5 days to assess response in shorter and longer time points. Expression of four selected miRNAs was measured: miR-143-3p, miR-107, miR-379-5p and miR-361-5p. These four miRNAs were chosen based on their novelty in cartilage biology as there is limited data on their role in OA. At the same time, we assessed expression of selected OA-related genes, collagen type II alpha 1 chain (*COL2A1*) aggrecan (*ACAN*), matrix metallopeptidase 13 (*MMP13*) and a disintegrin and metalloproteinase with thrombospondin motifs 4 (*ADAMTS4*), to assess whether treated cells responded to IL-1β treatment.

For the 24 h time point, analysis identified that miR-107 and miR-361-5p had lower expression in the treated group but this did not reach significance. There was a trend (*p* = 0.06) for higher expression following IL-1β treatment for miR-143-3p. However, expression of miR-379-5p did not change. There was no change in *COL2A1* and *ACAN* expression following IL-1β treatment. In contrast, *MMP13* and *ADAMTS4* had significantly higher expression following treatment when compared to control (Figure 5).

Treatment of chondrocytes with 10 ng/mL IL-1β for 5 days resulted in a miRNA expression profile that more resembled findings from our microarray experiment. There were significant reductions in expression of miR-107, miR-361-5p and miR-379-5p following IL-1β treatment and a trend for reduced expression of miR-143-3p. Additionally, *COL2A1* and *ACAN* had significantly reduced expression and catabolic markers while *MMP13* and *ADAMTS4* had significantly increased expression in the treated group compared to control (Figure 6).

### 3.5. Expression of Selected miR-107 Target Genes in Human Primary OA Chondrocytes Following Mimics and Inhibitor Treatment and in Human Cartilage Tissue

Of the DE miRNAs that we validated with qPCR, we selected miR-107 and miR-143-3p for further analysis due to their novelty in cartilage biology, large fold change observed in the microarrays and interesting potential target genes. To investigate the role miR-107 in OA cartilage, we transfected OA chondrocytes treated with 10 ng/mL IL-1β with miRNA-specific mimics and inhibitors. Wingless-Related Integration Site 4 (*WNT4*) and Indian Hedgehog (*IHH*) were identified as potential targets of miR-107 through available online bioinformatic tools, including TargetScan [45], miRWalk [46], miRmap [47] and miRTar [48]. miR-107 has previously been associated with chondrocyte proliferation and apoptosis [49,50,51]. In addition, WNT signalling and IHH signalling have also been linked to chondrocyte apoptosis as these pathways are activated in OA and promote cartilage degradation [52,53]. Therefore, we wanted to explore their relationship with miR-107. qPCR analysis of treated OA chondrocytes demonstrated significantly lower *WNT4* and *IHH* expression in the mimic group compared to control and significantly higher expression in the inhibitor group compared to control (Figure 7).

To further investigate whether these genes were targeted by miR-107, we interrogated their expression in an independent cohort of cartilage tissue samples that were used to validate microarray findings. Results for *WNT4* and *IHH* showed that these genes had significantly higher expression in old OA intact and lesioned cartilage tissue from OA patients compared to young intact cartilage from ACL donors (Figure 8). As shown above, miR-107 had lower expression in old OA intact and lesioned cartilage tissue compared to young intact cartilage tissue.

### 3.6. Proteomic Investigation of Human Primary OA Chondrocytes Treated with miR-143-3p Mimic and Inhibitor

To investigate the role of miR-143-3p in cartilage and OA, we undertook a different approach compared to that for miR-107. We undertook an unbiased proteomic approach enabling interrogation of many predicted targets and associated pathways. miR-143-3p was one of the most dysregulated miRNAs in our microarray analysis. In addition, previous studies from our group, have identified miR-143 as a DE miRNA in OA [28,54]. Therefore, we were interested in further exploring its role.

A total of 2252 proteins were identified, 1628 with two or more peptides. We identified 87 DE proteins at *p* < 0.05 in the mimic/control comparison and 50 DE proteins at *p* < 0.05 in the inhibitor/control group (Appendix A). When we adjusted for multiple testing, we did not identify any significant DE proteins in the two comparisons (FDR-adjusted *p* < 0.05). Therefore, we continued our analysis with proteins that were significant at *p* < 0.05. Even though we did not identify common DE proteins between the two comparisons, the opposite direction of expression would suggest that these DE proteins could be potential targets of miR-143-3p. We uploaded the two sets into STRING, to check whether these proteins were interacting with each other. STRING analysis showed that, for each comparison, most proteins were strongly interacting with each other and were forming specific networks (Figure 9). Of interest is that the proteins forming these networks were ribosomal proteins and translation initiation factors.

We hypothesised that DE proteins might be part of specific pathways regulated by miR-143-3p through targeting of genes that were not identified using LC-MS/MS. To investigate this hypothesis further, we uploaded the list of identified proteins for each comparison in IPA and carried out a ‘Core Analysis’ for each list, setting a *p* value cut-off at 0.05. Core Analysis showed that there were no common DE proteins between the two lists. For both mimic vs. control and inhibitor vs. control, the top canonical pathway identified was EIF2 signalling (Figure 10). In fact, in the mimic vs. control comparison, EIF2 signalling had a negative z-score [z-score= −1, −log(*p*-value) = 6.72], whereas, in the inhibitor vs. control comparison, EIF2 signalling had a positive z-score [z-score = 2, −log(*p*-value) = 7.54]. This suggests activation of EIF2 signalling in the inhibitor group and inhibition in the mimic group.

Next, we interrogated the upstream regulators of the pathways identified using IPA for each comparison. In the mimic vs. control comparison, La ribonucleoprotein 1 translational regulator (LARP1) was predicted to be an activated upstream regulator (activation z-score = 2.12). Interestingly, when investigating inhibitor vs. control, LARP1 was identified as being an inhibited upstream regulator (activation score z = −2.83) (Figure 11).

## 4. Discussion

MiRNAs play important roles in maintaining homeostasis, and dysfunctional miRNA networks could contribute to disease and vice versa [9,55]. There is significant research on the role of miRNAs in OA [56,57]. Many are implicated in OA, either having a protective or a destructive role in cartilage and usually linked to important cellular processes, including inflammation, autophagy and senescence [25]. In the present work, we defined the miRNA signatures of ageing and OA knee cartilage and investigated the role and targetome of specific miRNAs in cartilage ageing and disease. Our microarray analysis was undertaken on young intact cartilage and old OA cartilage. Young cartilage was sampled from donors undergoing ACL reconstruction. Time of surgery ranged from weeks to a few months following rupture. Since ACL rupture is a risk factor for post-traumatic OA, we chose to refer to this group as ‘Young intact’ cartilage rather than ‘Young healthy’. Studies have shown that 30% of patients who have ACL reconstruction surgery, have signs of cartilage degeneration, evident by Kellgren–Lawrence scoring (Score = 1 or 2), ten years post-surgery [58]. However, such alterations seem to appear years after reconstruction, therefore, we considered young intact cartilage as non-OA. It would have been beneficial to assess these tissues histologically and confirm integrity of the tissue, but given the limited size of cartilage tissue collected during these surgeries, this was not possible. In contrast, for OA samples, both Kellgren–Lawrence scores and histological data were available. As expected, lateral condyles were graded significantly lower than medial condyles using the Kellgren–Lawrence scoring system [59]. In addition, whilst intact samples appeared less damaged, the difference did not reach statistical significance. This may be due to donor variability and because cartilage biopsies were taken from specific areas of the condyles that might not be representative of the joint as a whole [60].

Microarray analysis between young intact, old OA intact and old OA lesioned samples identified significant changes in the expression of multiple miRNAs between groups. There were at least two variables that drove those changes: ageing and OA. We hypothesise that these changes were driven mainly by disease. Previous work from our group in young and old *equine* chondrocytes from donors without OA and macroscopically normal joints identified five DE miRNAs. Whilst a direct comparison cannot be made, given species differences, starting material (cartilage tissue versus chondrocytes), and experimental approaches (qPCR versus small RNA-Seq), in both experiments we observed moderate miRNA expression changes with ageing. Though limited in number, other studies which have interrogated the expression of miRNAs in cartilage ageing without the effect of OA, support our findings [61].

The majority of DE miRNAs had lower expression in the old OA cartilage groups compared to the young intact group. This could be due to disruption of the miRNA processing machinery during OA. Studies have shown that expression of DGCR8, part of the microprocessor complex, decreased with ageing in human MSCs, and overexpression of DGCR8 alleviated cartilage degradation in mice with ACL-induced OA [62]. Similarly, genetic deletion of DICER and DROSHA led to increased numbers of hypertrophic chondrocytes, increased chondrocyte death and decreased chondrocyte proliferation and OA-like symptoms in *murine* joints [15,63]. These studies suggest that changes in the cellular components involved in miRNA biogenesis could lead to decreased miRNA processing and, therefore, reduced miRNA expression.

Following microarray analysis, we were able to verify our results using qPCR quantification both in the same sample cohort and in additional cartilage samples. Of the validated DE miRNAs, four were chosen for follow up analysis: miR-361-5p, -379-5p, -107 and -143-3p. These miRNAs were chosen based on their novelty in cartilage biology as there is limited data on their role in OA. All four miRNAs had lower expression in human primary OA chondrocytes treated with IL-1β, one of the major pro-inflammatory cytokines. IL-1β treatment of primary chondrocytes is a common approach in miRNA OA research [64,65,66]. Microarray analyses in OA primary chondrocytes stimulated with IL-1β, revealed that most miRNAs which responded were downregulated [67,68], thus consistent with our findings. Disruption of the miRNA processing machinery could lead to decreased expression of miRNAs in disease and it is possible that inflammation in the joint could also contribute to it. Studies have demonstrated that overexpression of components involved in miRNA biogenesis, such as DGCR8, reduced the level of inflammatory markers, including IL-1β and IL-6 [62]. However, it is also important to note that the duration of treatment can affect miRNA response. In the 24 h timepoint, none of the miRNAs had significantly altered expression, although specific trends were observed for both miR-361-5p and miR-107. In contrast, in the 5-day timepoint, miR-361-5p, miR-107 and miR-379-5p had significant expression changes. MiR-143-3p did not show changes in both timepoints tested, however, opposite expression trends were observed following 24 h and 5 days of IL-1β treatment. Ji et al. (2016) undertook a similar experiment studying miR-30a in OA chondrocytes treated with 10 ng/mL and observed larger changes in miRNA expression at the 24 h time point compared to the 6 h time point [69]. Moreover, in a similar experiment, OA chondrocytes treated with 10 ng/mL IL-1β observed greater changes in expression of miR-92a-3p after 24 h of treatment compared to 6 h [70]. There are no studies which measured miRNA expression profiles in OA chondrocytes treated with IL-1β for a longer period. Most studies are conducted for shorter time points of up to 24 h [66,70,71]. With that said, we assessed chondrocyte responsiveness to IL-1β treatment, by measuring the expression of specific cartilage markers both at the 24 h and 5-day time point. Some studies have looked at the expression of specific OA markers, such as collagens and MMPs after treatment for longer periods. Lee et al. (2002) treated OA chondrocytes with IL-1β for 14 days and reported increased expression of *MMP13*, starting from the earlier time points and up to 14 days in the treated group compared to control [72]. In another study, bovine cartilage explants treated for 24 days demonstrated increased collagen and glycosaminoglycan content loss in the treated group [73]. In the same study, Lv et al. (2019) undertook RNA sequencing to assess differential expression of extracted chondrocytes treated with IL-1β for 48 h. *COL2A1* and *ACAN* had decreased expression and *MMP13* and *ADMATS4* increased expression following treatment [73]. In our experiment, *MMP13* and *ADAMTS4* had increased expression at both the 24 h and 5-day time points. *COL2A1* and *ACAN* did not show changes at the 24 h time point but had significant changes at the 5-day time point. This shows the complex regulatory mechanisms that characterise the inflammatory response in chondrocytes.

To further investigate the role of the selected miRNAs, we undertook overexpression and knock-down experiments for two miRNAs, miR-107 and miR-143-3p. Of the four miRNAs selected for further evaluation, miR-143-3p showed the highest fold change in expression between young intact and OA cartilage in the microarray experiment. miR-107 also showed a high fold change and we were interested in further exploring its role in chondrocyte proliferation. The rationale behind the addition of IL-1β in chondrocytes treated with mimics or inhibitors was to evaluate the benefit of these agents when chondrocytes were in an inflammatory environment, and whether these could potentially be used to treat patients in vivo [74,75].

Previous studies have shown that miR-107 has reduced expression in OA cartilage tissue and chondrocytes compared to normal controls, which is in agreement with our study [49,50,51]. These studies described that miR-107 promoted chondrocyte proliferation, and inhibition of its expression lead to apoptosis. It was shown that this miR-107 function was achieved by targeting genes involved in cell survival and proliferation and autophagy, such as TNF Receptor-Associated Factor 3 (*TRAF3*), phosphatase and tensin homolog (*PTEN*), and caspase 1 (*CASP1*). Based on the published data for miR-107, we decided to further explore the role of this miRNA in chondrocyte proliferation. For this, we focused on specific predicted miR-107 targets, *WNT4* and *IHH*. In our study, we found that *WNT4* and *IHH* could be potential target genes of miR-107. Analysis, showed that *WNT4* and *IHH* had reduced expression in chondrocytes treated with miR-107 mimic and increased expression when treated with the inhibitor, supporting the hypothesis that miR-107 could target these genes. Expression of miR-107 was not tested in these experiments, because according to manufacturer’s recommendations, qPCR for the miRNA itself will not confirm successful overexpression or inhibition of the miRNA of interest as qPCR amplification cannot distinguish between functional and non-functional mimics and inhibitors [76,77]. In addition, these synthetic molecules could interfere with PCR amplification [76,77]. The problematic use of qPCR amplification of the miRNA of interest to assess successful overexpression and inhibition has also been discussed by others [78]. Therefore, we focused on the expression of the potential target genes.

To further assess the potential targeting of *WNT4* and *IHH* by miR-107, we measured their expression in OA intact/lesioned cartilage and young intact cartilage from our independent cohort of donors. *WNT4* and *IHH* had increased expression in OA intact/lesioned cartilage compared to young intact cartilage, opposite to the expression of miR-107 indicating that miR-107 could indeed target *WNT4* and *IHH*. The role of *WNT4* in OA is still unclear. Studies have shown that *WNT4* was overexpressed in human OA hip and knee cartilage [79], and in mice with induced-OA [80]. In the latter study, induced-OA led to increased *WNT4* and activated the WNT signalling pathway, leading to increased expression of MMP2 and MMP9 through β-catenin. In a more recent study by Yao et al. (2021), the authors reported that Bushen Qiangjin capsule (BSQJ), a medicine used in Chinese tradition, could attenuate papain-induced OA in rat by inhibiting the WNT/a-catenin pathway. In this study, *WNT4* had increased expression in rats with induced OA and was downregulated after treatment, which indicates that *WNT4* is involved in cartilage degradation [81]. The role of *IHH* in cartilage biology is better researched as it is involved in chondrocyte hypertrophy during development [82]. Studies have shown that the IHH signalling is dysregulated in OA and *IHH* expression is elevated in OA cartilage to promote hypertrophy, apoptosis and senescence [83,84,85]. In another study, inhibition of IHH signalling by ipriflavone, a compound that inhibits the IHH pathway, led to increased chondrocyte proliferation and reduced apoptosis and cartilage degeneration [53]. Given the biological pathways that *WNT4* and *IHH* are involved in, our work indicates that, miR-107 could also be involved in chondrocyte proliferation, hypertrophy and apoptosis via the targeting of these genes.

To investigate the role of miR-143-3p in cartilage and OA we undertook a different approach than miR-107. As miR-143-3p was one of the most differentially expressed miRNAs in our microarray dataset, we were interested in exploring its effect on the chondrocyte proteome and its wider role/function in cartilage biology rather than focusing on specific targets genes. To evaluate the effect of miR-143-3p overexpression/inhibition on the chondrocyte proteome, we used LC-MS/MS between miR-143-3p mimic vs. control, and inhibitor vs. control, and identified DE proteins. There were no common DE proteins between the two comparisons that would identify specific targets of miR-143-3p. However, using a bioinformatic approach we identified that both sets of DE proteins were involved in EIF2 signalling. The potential regulation of EIF2 signalling could be through specific miR-143-3p target genes not identified using LC-MS/MS, either because their detection was outside of instrument sensitivity or because they act at an earlier time point, but their results can still be observed several hours and until collection of lysates, that is 48 h after transfection. Either way, our results provided a potential role of miR-143-3p and its association in EIF2 signalling that requires further investigation. EIF2 signalling is involved in protein translation and synthesis and plays a crucial role in cellular response due to stress-related stimuli, such as oxidative stress, injury, and others [86]. Under stress stimuli, EIF2 is phosphorylated, leading to its inhibition and reduced global protein translation. Instead, specific genes are expressed which promote cellular adaptation and survival and exert an anti-apoptotic effect. Interestingly, in a study assessing the transcriptomic differences between ovaries from young and middle-aged mice, the authors reported increased miR-143 expression and reduced EIF2 signalling in middle-aged ovaries compared to young [87]. Even though the authors did not discuss the potential association of miR-143 to EIF2 signalling, our results suggest a potential link between miR-143-3p and EIF2. miR-143-3p could be involved in cartilage homeostasis through regulating EIF2 signalling. Specifically, overexpression of miR-143-3p could inhibit EIF2 signalling, leading to repression of global protein synthesis, but promoting the expression of genes that assist chondrocyte survival during stress. In contrast, inhibition of miR-143-3p during OA (as suggested by our microarray study), could induce EIF2 signalling, which, under stress conditions, could result in depletion of cellular components important for cell survival, thus leading to increased apoptosis.

Additionally, we identified upstream regulators of the DE proteins for each comparison. Amongst these, LARP1 was a common upstream regulator in both comparisons predicted to be activated in mimic-treated chondrocytes, and inhibited in chondrocytes treated with the inhibitor. LARP1 is involved in protein translation, cell proliferation and cell survival and has been shown to interact with ribosomal proteins. It is part of the mTOR signalling pathway, suggesting a role of LARP1 in miR-143-3p-related pathways [88]. Though there are contradictory data on its role, it is believed that LARP1 inhibits the translation of ribosomal proteins and translation factors [89]. However, further studies are needed to elucidate its exact role, especially with regard to cartilage biology, as there are currently no data on its role in OA.

Taken together, our results elucidate the complex molecular landscape that characterises OA pathogenesis and support the implication of miRNAs in cartilage biology and disease. The large number of miRNAs that were DE in our microarray data highlight further dysregulation of multiple molecular pathways in knee OA, but, at the same time, provide several potential therapeutic targets for additional research. The clustering of OA samples into subgroups, indicates distinctive miRNA expression profiles amongst OA patients. This clustering has also been observed previously for protein-coding genes in OA patients [90]. OA is a heterogeneous disease characterised by different phenotypes, risk factors and clinical symptoms amongst patients [91]. Different phenotypes of OA including inflammatory OA, metabolic OA, biomechanical OA and osteoporotic OA, have been proposed [91]. In our dataset, we showed that longer exposure of chondrocytes to IL-1β can have an aberrant effect on miRNA expression. Taking into account that subgrouping in our dataset did not correlate with radiographic and histological scoring, this suggests that an “one size fits all” strategy might not be suitable in OA therapeutics and personalised approaches should be considered in the future accounting for unique molecular signatures and disease characteristics.

In the current study, we were able to investigate further the role of specific miRNAs in OA cartilage. While previous studies have provided evidence on the role of miR-107 in chondrocyte proliferation and apoptosis by targeting genes such as *PTEN* and *TRAF3* [50,51], to the best of our knowledge, this is the first study to suggest that miR-107 could also target *IHH* and *WNT4*. Since these genes are involved in chondrocyte hypertrophy and apoptosis, our results further support the involvement of miR-107 in chondrocyte proliferation, which is in line with previous reports, and also reinstate the fact that miRNAs act by targeting multiple mRNAs to modulate gene expression. This highlights the need to better understand the interactions of these molecules in order to design suitable therapeutic agents. This is also true for miR-143-3p and EIF2 signalling. There is limited information regarding the role of miR-143-3p in OA or the role of EIF2 signalling in OA. One study investigated the effect of curcumin on miR-143-3p in an OA mouse model [92], whereas another study investigated the effect of curcumin on EIF2 signalling in an OA rat model [93]. The first study reported a positive effect of curcumin on miR-143-3p expression which attenuated OA-like changes, whereas the second study reported an inhibitory effect of curcumin on EIF2 signalling which also attenuated OA-like changes. While there is no association made between miR-143-3p and EIF2 signalling in either study, our results indicate that there could be a potential link between miR-143-3p and EIF2 signalling in cartilage and OA. To the best of our knowledge, this has not been reported previously.

In this study, we aimed to investigate the role of miRNAs in cartilage and OA. While our findings have shed light into the potential mechanisms involved in OA pathogenesis, there are certain limitations in our experimental design and approach. In our microarray experiment, we used young intact cartilage samples from ACL patients and OA samples from older patients undergoing TKA. It would have been beneficiary to include aged-matched normal cartilage samples to separate the age effect from disease. However, as normal cartilage samples are very difficult to obtain, this was not possible. Moreover, we have used an in vitro OA-like inflammatory model to assess changes in miRNA expression in OA primary chondrocytes. With that said, it is important to consider additional factors that contribute to OA, such as mechanical stress, injury, obesity, genetics, joint structure and others. Even though this in vitro OA inflammatory model is commonly used in miRNA research in OA, it does not fully represent the underlying molecular landscape of OA. Additional experiments involving mechanotransduction and 3D cultures could shed more light into the role and function of these miRNAs. Furthermore, additional experiments are needed to characterise sufficiently the relationship between miR-107 and *WNT4* and *IHH*. Experimental approaches, such as Western blotting and luciferase assays, would have confirmed whether miR-107 directly targets *WNT4* and *IHH*. Additional knock-down experiments would have provided more information on the role of *WNT4* and *IHH* in OA and their potential interaction with miR-107. The same applies to miR-143 and EIF2 signalling. Additional experiments are needed to confirm their relationship and their overall implication in OA pathogenesis. Nonetheless, our experiments have set the basis for further exploration of molecular pathways in OA and have generated a substantial amount of data for future research in this field.

## 5. Conclusions

Our analyses have pointed to the complex mechanistic networks that underlie cartilage pathology, thus laying the ground for further research in this field. Two miRNAs of interest, miR-107 and miR-143-3p, have potential implication with key molecular pathways associated with chondrocyte proliferation and survival.

## Figures and Tables

**Figure 1 biomedicines-11-01189-f001:**
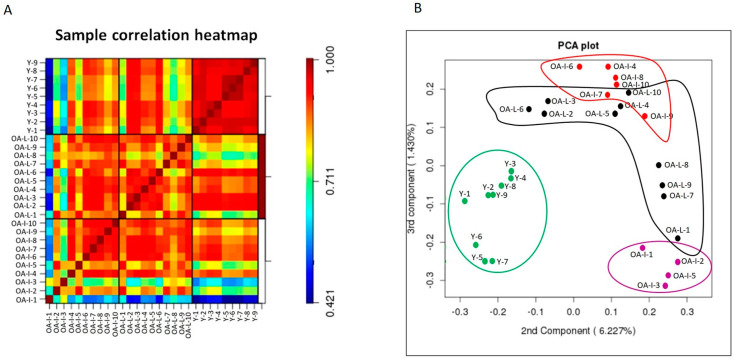
Microarray data visualisation. (**A**). Heatmap of sample correlation coefficients. Correlation coefficients following microarray analysis between young intact, old OA intact and lesioned cartilage samples were computed using log-transformed expression values of detected miRNAs. Colour gradient from blue to red denotes transition from low to high correlation coefficients. Heatmap generated in R. (**B**) 2D PCA plot of the second and third components of miRNA abundance in young intact, old OA intact and old OA lesioned samples. Clustering of young intact (Y), old OA intact (OA-I) and lesioned (OA-L) cartilage samples based on miRNA expression, following microarray analysis. Green: Young intact, magenta: old OA-Intact-1, red: old OA-Intact-2, black: old OA-Lesioned. Numbers (1 to 9 for young intact and 1 to 10 for old OA intact/lesioned) correspond to donor/patient number. PCA plot generated in R.

**Figure 2 biomedicines-11-01189-f002:**
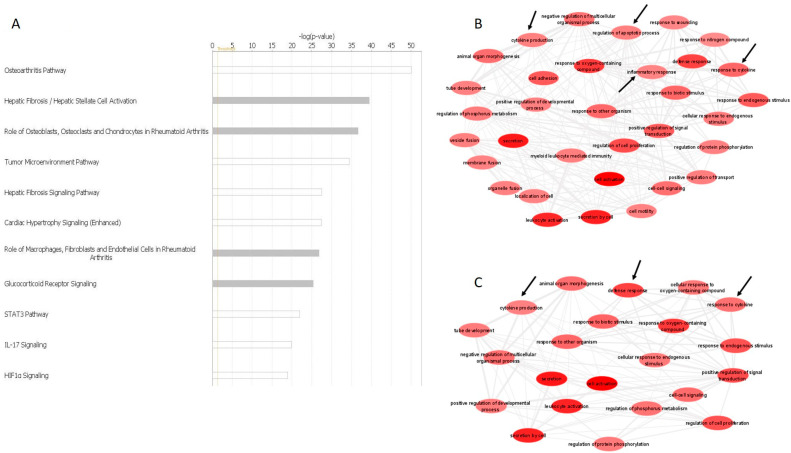
Bioinformatics of DE miRNAs. (**A**) Top canonical pathways of validated and predicted mRNA targets of DE miRNAs in human cartilage samples. Core analysis of highly predicted mRNA targets of DE miRNAs in OA-Lesioned vs. Y, OA-Intact-1 vs. Y and OA-Intact-2 vs. Y, revealed the most important pathways involving these mRNAs. Pathways were ranked based on level of significance [−log(*p*-value)], dependent on the number of mRNAs involved in each pathway. The figure depicts biological pathways generated from validated and predicted mRNA targets of the DE miRNAs identified in the OA-Lesioned vs. Y comparison. DE miRNAs in OA-Intact-1 vs. Y and OA-Intact-2 vs. Y comparisons were similar and, therefore, top biological pathways were also similar. Graph generated in IPA Version 60467501. (**B**,**C**) Pathway interaction analysis of predicted targets of DE miRNAs from comparisons OA-Intact-1 vs. Y and OA-Intact-2 vs. Y. Predicted target genes of DE miRNAs from comparisons (**B**) OA-Intact-1 vs. Y and (**C**) OA-Intact-2 vs. Y were uploaded to ToppGene Suite which generated a list of GO terms. GO terms were pasted in Revigo for selection of key biological pathways. Biological pathways were visualised using Cytoscape Version 3.8.2, which generated the figures. Important OA-related pathways are indicated with black arrows.

**Figure 3 biomedicines-11-01189-f003:**
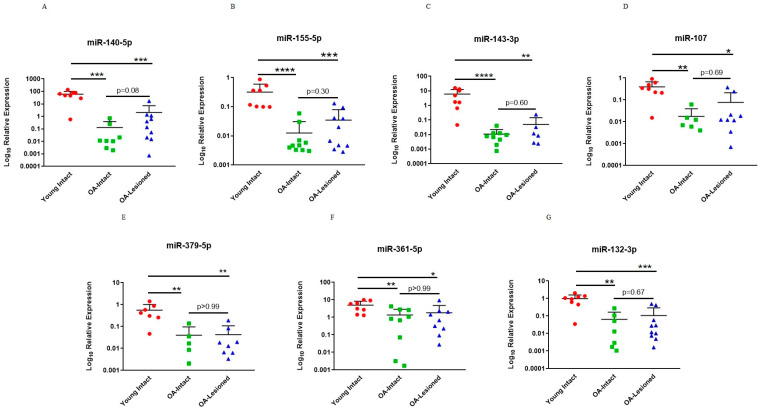
qPCR validation of selected DE miRNAs from the dependent cohort. Seven miRNAs; (**A**) miR-140-5p, (**B**) miR-155-5p, (**C**) miR-143-3p, (**D**) miR-107, (**E**) miR-379-5p, (**F**) miR-361-5p and (**G**) miR-132-3p, which were DE in microarray analysis were selected for validation with qPCR in the same samples. *n* = 5–10 samples/group. Y axis represents log-transformed expression relative to miR-6786-5p. Statistical analysis was undertaken using a Mann–Whitney test in GraphPad Prism (Version 8.0.1). Data are represented as mean + SD. Red: Young group, Green: OA-Intact group, Blue: OA-Lesioned group. *p* values < 0.05 were considered significant. *: *p* < 0.05, **: *p* < 0.01, ***: *p* < 0.001, ****: *p* < 0.0001.

**Figure 4 biomedicines-11-01189-f004:**
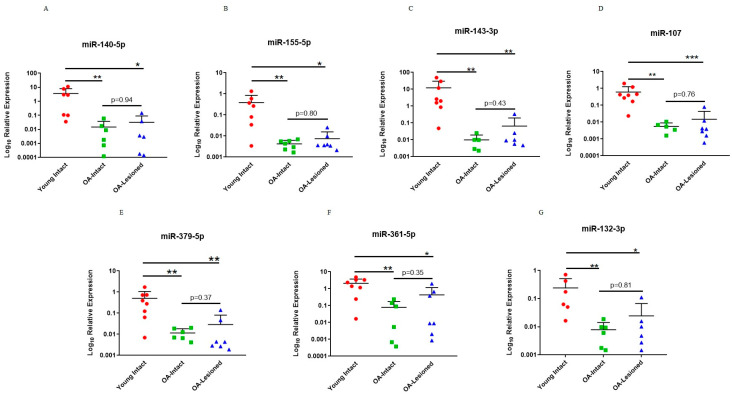
qPCR validation of selected DE miRNAs in the independent cohort. Seven miRNAs; (**A**) miR-140-5p, (**B**) miR-155-5p, (**C**) miR-143-3p, (**D**) miR-107, (**E**) miR-379-5p, (**F**) miR-361-5p and (**G**) miR-132-3p which were DE in microarray analysis were selected for validation with qPCR in an independent cohort of cartilage samples. *n* = 5–10 samples/group. Y axis represents log-transformed expression relative to miR-6786-5p. Statistical analysis was undertaken using a Mann–Whitney test in GraphPad Prism (Version 8.0.1). Data are represented as mean + SD. Red: Young group, Green: OA-Intact group, Blue: OA-Lesioned group. *p* values < 0.05 were considered significant. *: *p* < 0.05, **: *p* < 0.01, ***: *p* < 0.001.

**Figure 5 biomedicines-11-01189-f005:**
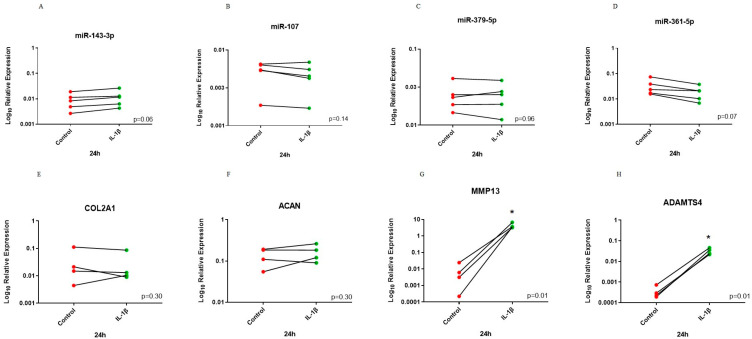
Before–after plot showing expression of selected miRNAs and cartilage markers in human OA chondrocytes treated with IL-1β for 24 h. Human OA chondrocytes were treated with 10 ng/mL human recombinant IL-1β or control for 24 h and expression of selected miRNAs; (**A**) miR-143-3p, (**B**) miR-107, (**C**) miR-379-5p and (**D**) miR-361-5p and cartilage markers; (**E**) *COL2A1*, (**F**) *ACAN*, (**G**) *MMP13* and (**H**) *ADMATS4* was measured (*n* = 4–5 donors). Y axis represents log-transformed expression relative to SNORD68 for miRNAs and relative to *GAPDH* for OA markers. Statistical analysis was undertaken using a two-tailed paired t-test in GraphPad Prism (Version 8.0.1). *p* values < 0.05 were considered significant. *: *p* < 0.05. Each line connects the control (red) and treated group (green) of the same donor.

**Figure 6 biomedicines-11-01189-f006:**
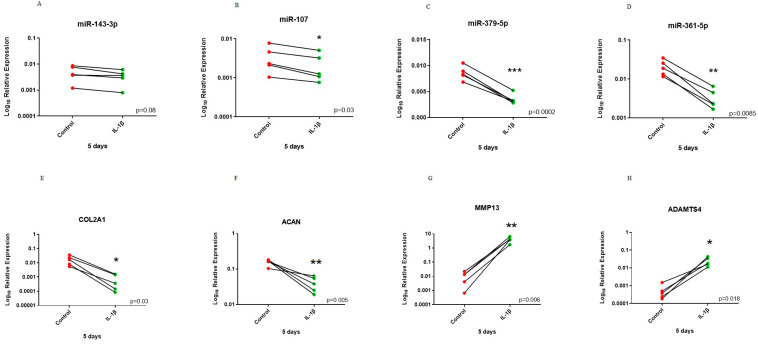
Before–after plot showing expression of selected miRNAs and cartilage markers in human OA chondrocytes treated with 10 ng/mL IL-1β for 5 days. Human OA chondrocytes were treated with 10 ng/mL human recombinant IL-1β for 5 days and expression of selected miRNAs; (**A**) miR-143-3p, (**B**) miR-107, (**C**) miR-379-5p and (**D**) miR-361-5p and cartilage markers; (**E**) *COL2A1*, (**F**) *ACAN*, (**G**) *MMP13* and (**H**) *ADMATS4* was measured (*n* = 5 donors). Y axis represents log-transformed expression relative to SNORD68 for miRNAs and relative to *GAPDH* for OA markers. Statistical analysis was undertaken using a two-tailed paired t-test in GraphPad Prism (Version 8.0.1). *p* values < 0.05 were considered significant. *: *p* < 0.05, **: *p* < 0.01, ***: *p* < 0.001. Each line connects the control (red) and treated group (green) of the same donor.

**Figure 7 biomedicines-11-01189-f007:**
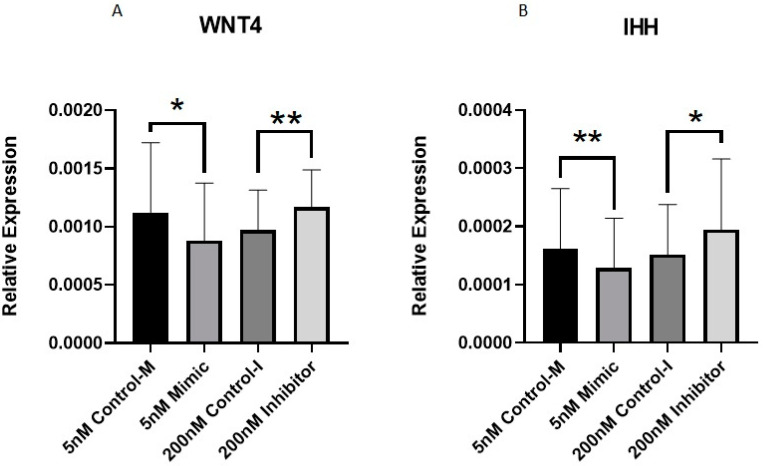
Expression of *WNT4* and *IHH* in human primary OA chondrocytes treated with miR-107 mimic, inhibitor or control. Human OA chondrocytes were treated in media supplemented with 10 ng/mL IL-1β with 5 nM miR-107 mimic, 200 nM miR-107 inhibitor or control for 48 h. Expression of (**A**) *WNT4* and (**B**) *IHH* are shown (*n* = 8–11 donors). Expression is relative to *GAPDH*. Statistical analysis was undertaken using a two-tailed paired t-test in GraphPad Prism (Version 8.0.1). Data are represented as mean + SD. *p* values < 0.05 were considered significant. *: *p* < 0.05, **: *p* < 0.01. Control m = control-mimic, control-I = control-inhibitor.

**Figure 8 biomedicines-11-01189-f008:**
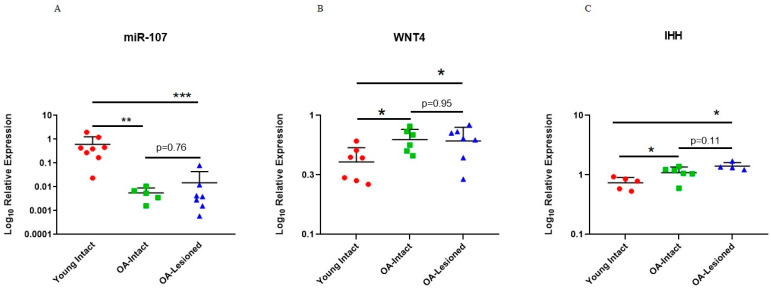
Expression of miR-107, *WNT4* and *IHH* in the independent cohort of young intact and old OA intact and lesioned cartilage samples. Expression of (**A**) miR-107, (**B**) *WNT4* and (**C**) *IHH* in young intact cartilage from ACL donors and old OA intact and lesioned cartilage is shown. Expression of miR-107 from Figure 4 is shown here again for direct comparison (*n* = 5–8 donors). Y axis represents log-transformed expression relative to SNORD68 for miR-107, and relative to *RPL3* for *WNT4* and *IHH*. Statistical analysis was undertaken using a Mann–Whitney test in GraphPad Prism (Version 8.0.1). Data are represented as mean + SD. *p* values < 0.05 were considered significant. *: *p* < 0.05, **: *p* < 0.01, ***: *p* < 0.001.

**Figure 9 biomedicines-11-01189-f009:**
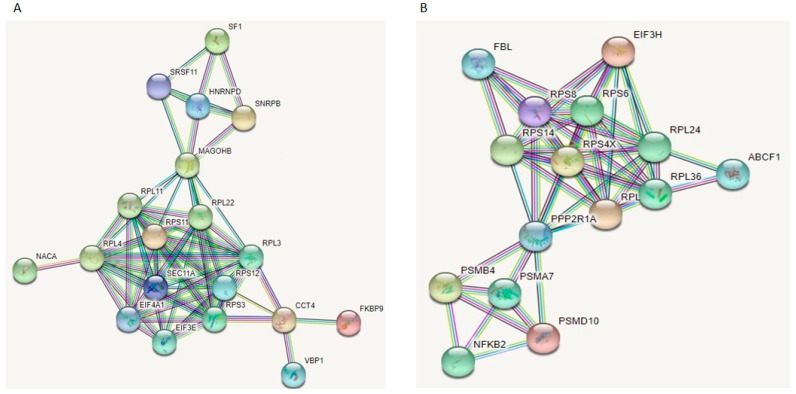
STRING analysis of DE proteins identified in chondrocytes treated with miR-143-3p mimic, inhibitor or control following LC-MS/MS. DE proteins (*p* < 0.05) between (**A**) mimic vs. control group and (**B**) inhibitor vs. control group, were uploaded on STRING for protein network analysis and to identify protein–protein interactions. Filters were set to show both functional and physical protein associations, and highest confidence (0.900) was selected. Line colour indicates the type of interaction evidence. Stronger associations are represented by thicker lines. Figure generated with STRING.

**Figure 10 biomedicines-11-01189-f010:**
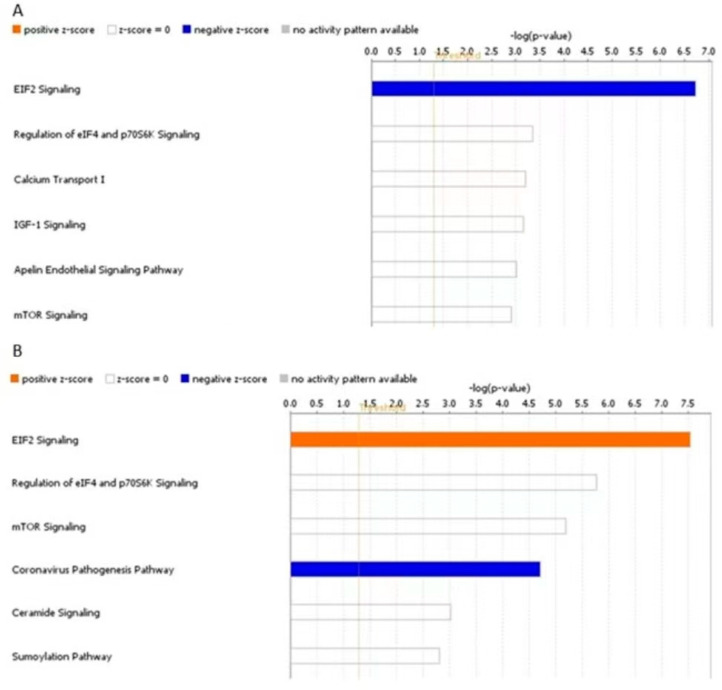
Top canonical pathways linked to DE proteins from human OA chondrocytes treated with miR-143-3p mimic, inhibitor or control. Human OA chondrocytes were treated in media supplemented with 10 ng/mL IL-1β with miR-143-3p mimic, inhibitor or control for 48 h. Cell lysates were collected and subjected to LC-MS/MS. Figure depicts IPA Core Analysis of DE proteins from chondrocytes treated with (**A**) miR-143-3p mimic or control and (**B**) miR-143-3p inhibitor or control. Top canonical pathways associated with each dataset and z activation scores for each canonical pathway are shown. Figures generated with IPA Version 60467501.

**Figure 11 biomedicines-11-01189-f011:**
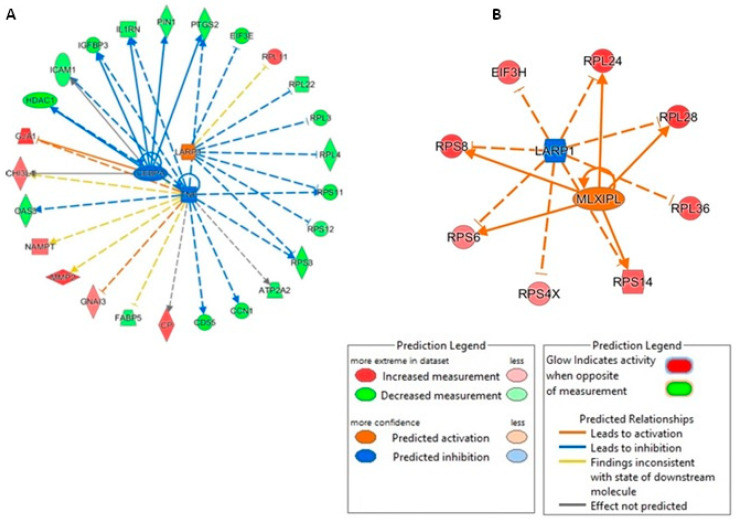
Network analysis of predicted upstream regulators of DE proteins in human OA chondrocytes treated with miR-143-3p mimic, inhibitor or control. IPA analysis of DE proteins (*p* < 0.05) in human OA chondrocytes treated with miR-143-3p (**A**) mimic or control and (**B**) inhibitor or control, identified a set of predicted upstream regulators in each comparison. Figure generated with IPA. Key is indicated.

**Table 1 biomedicines-11-01189-t001:** Number of significant DE miRNAs in human cartilage samples. Significant DE miRNAs are defined as those with FDR-adjusted *p*-value < 5%. OA-Lesioned/OA-Intact 1/OA-Intact-2: group of interest. Young: reference group.

	OA-Lesioned vs. Young	OA-Intact-1 vs. Young	OA-Intact-2 vs. Young
No of DE miRNAs with higher expression in the group of interest	20	69	19
No of DE miRNAs with lower expression in the group of interest	298	408	313
Total	318	477	332

OA—osteoarthritis, DE—differentially expressed.

## Data Availability

Data are deposited in NCBI’s Gene Expression Omnibus (GEO Series accession number E-MTAB-5716).

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
