# Peer review of "MicroRNA Signatures in Cartilage Ageing and Osteoarthritis"

_biomedicines, 2023, doi:10.3390/biomedicines11041189_

Round 1
Reviewer 1 Report
The manuscript biomedicines-2266545 entitled "MicroRNA signatures in cartilage ageing and osteoarthritis" can be reconsidered for publication in the “Biomedicines” after major revisions.
The topic is interesting and the results are well organized and promising, however some modifications are required to make the manuscript suitable for publication.
Introduction:
The second part of this section concerning miRNA, biosynthesis and function should be improved and better described. The relative references should be updated.
The role of miRNA in OA pathogenesis and chondrocyte homeostasis has not been well described. The most recent and updated papers (also reviews) relative to this topic need to be included….see for instance:
De Palma A, et al. Do MicroRNAs have a key epigenetic role in osteoarthritis and in mechanotransduction? Clin Exp Rheumatol. 2017;35(3):518-526.
Swingler TE, et al. The function of microRNAs in cartilage and osteoarthritis. Clin Exp Rheumatol. 2019; 37 Suppl 120(5):40-47.
Xie F, et al. Role of MicroRNA, LncRNA, and Exosomes in the Progression of Osteoarthritis: A Review of Recent Literature. Orthop Surg. 2020;12(3):708-716.
Furthermore, more updated published papers on miRNA involved in the regulation of chondrocytes metabolism, apoptosis and oxidative stress are available from the literature.
Methods:
Methods section needs to be re-written and implemented. This section should provide enough deep information to allow readers to repeat experiments. The authors should explain why they undertook each experiment, why the used the methods they used, and how they conducted each experiment.
Please improve.
Discussion:
The authors should deeper elucidate what all of the findings taken together mean, and what are the larger implications of the obtained results.
Please add the limitations of the study.
Please check the English style.
Author Response
Point 1: Introduction: The second part of this section concerning miRNA, biosynthesis and function should be improved and better described. The relative references should be updated.
Response 1: We have added more details regarding miRNA biosynthesis and function and at the same we tried to convey the main points and keep the text concise, in line with the general format of the introduction. We have updated the reference list used at this part of the text.
Point 2: The role of miRNA in OA pathogenesis and chondrocyte homeostasis has not been well described. The most recent and updated papers (also reviews) relative to this topic need to be included. Furthermore, more updated published papers on miRNA involved in the regulation of chondrocytes metabolism, apoptosis and oxidative stress are available from the literature.
Response 2: We have updated the reference list to include recent references from the available literature. We have discussed in more details the role of miRNAs in cartilage homeostasis OA pathogenesis and OA-related processes such as inflammation, autophagy, oxidative stress and senescence. We have included examples of specific miRNAs and their role in the above mechanisms from the current literature.
Point 3: Methods: Methods section needs to be re-written and implemented. This section should provide enough deep information to allow readers to repeat experiments. The authors should explain why they undertook each experiment, why the used the methods they used, and how they conducted each experiment. Please improve.
Response 3: Additional details were included in the Methods section. Specifically, the following sections were supplemented:
- Isolation of human primary chondrocytes from OA articular cartilage
- Histology
- RNA Isolation
- qPCR for miRNA/mRNA quantification
- Treatment of human primary chondrocytes with IL-1β
- Protein extraction, in-solution digestion and LC-MS/MS following overexpression and knockdown
For the justification of the method used please see Discussion. Specifically:
- for the use of qPCR amplification please see the following sentence in paragraph 4 of discussion: Following microarray analysis, we were able to verify our results by qPCR quantification both in the same sample cohort and in additional cartilage samples
- for the use of IL-1β to treat primary chondrocytes please see paragraph 4 in discussion: IL-1β treatment of primary chondrocytes is a common approach…
- for the use of mimics/inhibitors of miR-107 and miR-143-3p please see paragraph 5 in discussion: To investigate further the role of the selected miRNAs, we undertook overexpression and knock-down experiments for two miRNAs; miR-107 and miR-143-3p… chondrocyte proliferation
- for the use of LC-MS/MS please see paragraph 8 in discussion: To investigate the role of miR-143-3p in cartilage and OA…
Point 4: Discussion: The authors should deeper elucidate what all of the findings taken together mean, and what are the larger implications of the obtained results.
Response 4: We have discussed further the deeper meaning of our results and what these could mean for cartilage biology and OA. We have also provided the strengths of our study and findings which, to the best of our knowledge, have not been reported before. Please check paragraphs 10 (Taken together, our results elucidate…) and 11 (In the current study, we were able to...) in discussion.
Point 5: Please add the limitations of the study.
Response 5: We have added the limitations of our study. Please refer to last paragraph in discussion.
Reviewer 2 Report
This is an interesting study. Some minor points are enlisted below
1. Line 83: "K&L" please avoid abbreviation
2. Line 91: please briefly extend the isolation process from reference 21, and important information about manufacturer should be offered.
3. Line 160~161:"miR-107 and miR-143-3p"suggest shift to " recognized miRNA" or else, because from the text, "miR-107 and miR-143-3p" should be described in later results section behind in line 322~334.
4. The authors chose WNT4 and IHH as target genes based upon online bioinformatic tools. Why are the role of WNT4 and IHH in OA pathogenesis. ? Please shift or repeat a little bit of content from line 614 ~ 628 to line 419 behind reference 41 to justify your choice for WNT4 and IHH.
5. Why choose different approaches for miR-143-3p and miR-107 concerning their roles in cartilage and OA?
Author Response
Point 1. Line 83: "K&L" please avoid abbreviation
Response 1: We have amended the abbreviation and used the term “Kellgren-Lawrence” throughout the text.
Point 2: Please briefly extend the isolation process from reference 21, and important information about manufacturer should be offered.
Response 2: We have added more details in our methods section including section “Isolation of human primary chondrocytes from OA articular cartilage isolation”. We have also added details about manufacturers.
Point 3: Line 160~161:"miR-107 and miR-143-3p"suggest shift to " recognized miRNA" or else, because from the text, "miR-107 and miR-143-3p" should be described in later results section behind in line 322~334.
Response 3: Could you please clarify what section you are referring to? And what changes you are suggesting? Thank you.
Point 4: The authors chose WNT4 and IHH as target genes based upon online bioinformatic tools. Why are the role of WNT4 and IHH in OA pathogenesis? Please shift or repeat a little bit of content from line 614 ~ 628 to line 419 behind reference 41 to justify your choice for WNT4 and IHH.
Response 4: We have added a brief explanation in section “Expression of selected miR-107 target genes in human primary OA chondrocytes following mimics and inhibitor treatment and in human cartilage tissue” of why we chose to investigate further the relationship between miR-107 and WNT4 and IHH. In addition, we have added more data in paragraph 7 in discussion regarding the role of WNT4 and IHH in cartilage and OA.
Point 5: Why choose different approaches for miR-143-3p and miR-107 concerning their roles in cartilage and OA?
Response 5: Please refer to paragraph 8 in discussion (To investigate the role of miR-143-3p in cartilage and OA…) where we have added our justification for using different approaches for investigating the role of miR-107 and miR-143-3p.
Round 2
Reviewer 1 Report
The Authors modified the text as required
Author Response
We relied to these comments in our previous rebuttal.